# Spectroscopy Study of Albumin Interaction with Negatively Charged Liposome Membranes: Mutual Structural Effects of the Protein and the Bilayers

**DOI:** 10.3390/membranes12111031

**Published:** 2022-10-23

**Authors:** Daria Tretiakova, Maria Kobanenko, Irina Le-Deygen, Ivan Boldyrev, Elena Kudryashova, Natalia Onishchenko, Elena Vodovozova

**Affiliations:** 1Shemyakin–Ovchinnikov Institute of Bioorganic Chemistry, Russian Academy of Sciences, ul. Miklukho-Maklaya 16/10, 117997 Moscow, Russia; 2Department of Chemistry, Lomonosov Moscow State University, Leninskie Gory 1/3, 119991 Moscow, Russia

**Keywords:** protein adsorption, albumin, liposome–protein complexes, fluorescence spectroscopy, ATR-FTIR

## Abstract

Liposomes as drug carriers are usually injected into the systemic circulation where they are instantly exposed to plasma proteins. Liposome–protein interactions can affect both the stability of liposomes and the conformation of the associated protein leading to the altered biodistribution of the carrier. In this work, mutual effects of albumin and liposomal membrane in the course of the protein’s adsorption were examined in terms of quantity of bound protein, its structure, liposome membrane permeability, and changes in physicochemical characteristics of the liposomes. Fluorescence spectroscopy methods and Fourier transform infrared spectroscopy (ATR-FTIR), which provides information about specific groups in lipids involved in interaction with the protein, were used to monitor adsorption of albumin with liposomes based on egg phosphatidylcholine with various additives of negatively charged lipidic components, such as phosphatidylinositol, ganglioside GM_1_, or the acidic lipopeptide. Less than a dozen of the protein molecules were tightly bound to a liposome independently of bilayer composition, yet they had a detectable impact on the bilayer. Albumin conformational changes during adsorption were partially related to bilayer microhydrophobicity. Ganglioside GM_1_ showed preferable features for evading undesirable structural changes.

## 1. Introduction

Albumin is the most abundant serum protein with concentration of about 35–50 mg/mL [1]. It is a cargo protein for a diverse array of endogenous and exogenous small molecules [2] and it has several very high affinity binding sites for fatty acids [3]. Albumin is a highly flexible molecule. It undergoes conformational changes upon binding of ligands, which can result in both partial protein unfolding (increase of random coil content) leading to destabilization of the protein, or an increase in α-helical content, leading to the protein structure’s stabilization [4]. Serum albumin is also often found among the most abundant proteins adsorbed on nanosized drug delivery systems, such as liposomes, injected into blood flow. Initially, protein binding to the liposome surface is determined by the protein’s bulk concentration, but later on it can be exchanged based on the protein’s affinity to the bilayer (Vroman effect [5]). Proteins bind the membrane via electrostatic interactions, as well as hydrophobic interactions and H bonds. To create additional contacts with the surface, they are apt for orientational and conformational changes that increase their affinity for the bilayer.

In circulation, when a protein bound to a drug carrier changes conformation, it can alter biodistribution of the drug. For example, normally glycoprotein receptor gp60 translocates albumin across the endothelium to the interstitium. Yet other gp receptors can scavenge conformationally modified albumin and deliver the whole system (liposome-bound protein) to the vascular endothelium for further lysosomal degradation [4,6]. Fleischer et al. [7] have shown that if albumin binding to a nanoparticle surface requires changes in the protein secondary structure, then modified albumin is recognized by a scavenger receptor, SR-B1. Consequences of structural changes in various proteins upon adsorption onto nanoparticles have gained attention from researchers as changes in protein conformation may cause immunogenicity, aggregation, or loss of biocompatibility of the nanoparticles [8,9,10,11,12]. In particular, albumin can both prolong liposome circulation time [13] or induce macrophage clearance and trigger recognition by innate immunity [14,15].

In this work we investigated albumin interactions with fluid-phase liposomes that include one of the negatively charged amphiphilic molecules: phosphatidylinositol (PI), ganglioside GM_1_, or a carboxymethylated oligoglycine (CMG) lipid conjugate. All these molecules have been previously tested as stabilizing components [16] or shielding molecules [17] for liposomes carrying lipophilic anticancer prodrugs. Liposomes with such shielding molecules in the bilayer had some advantages compared to liposomes bearing only prodrugs. Still, we observed phagocytosis of these liposomes and release of the dye from the aqueous compartment. The aim of this study was to clarify if liposomes with PI, GM_1_, or carboxymethylated oligoglycine still bind albumin in the amount sufficient to affect membrane permeability and trigger undesirable conformational changes of the protein during adsorption.

## 2. Materials and Methods

### 2.1. Chemicals

Egg phosphatidylcholine (ePC; USP grade, Lipoid E PC S) was obtained from Lipoid GmbH (Heidelberg, Germany); raw soybean phosphatidylinositol (PI) was a kind gift from Lipoid, and it was further purified by column chromatography on silica gel and characterized by 1H NMR spectroscopy as an individual phospholipid; ganglioside GM_1_ from bovine brain was obtained from Sigma; 1,3,5,7-tetramethyl-BODIPY-labeled phosphatidylcholine (TMB-PC) was synthesized as previously reported [18]; carboxymethylated oligoglycine conjugate with dioleoylphosphatidylethanolamine (CMG-PE) previously used in our studies [16,17] was a kind gift from the Laboratory of Carbohydrates (Shemyakin–Ovchinnikov Institute of Bioorganic Chemistry, RAS). Calcein (tetrasodium salt) was purchased from Serva; 8-anilino-1-naphthalenesulfonic acid ammonium salt (ANS) was obtained from Fluka; Sephadex G-50 was obtained from Pharmacia. Bovine serum albumin (BSA) was purchased from Dia-m (Moscow, Russia).

Buffer compositions were as follows: phosphate buffered saline (PBS; KH_2_PO_4_, 0.2 g/L; NaH_2_PO_4_ × 2H_2_O, 0.15 g/L; Na_2_HPO_4_, 1.0 g/L; KCl, 0.2 g/L; NaCl, 8.0 g/L, pH 7.4); Tris-buffered saline (TBS; NaCl, 4.39 g; Tris, 3.03 g; H_2_Odd, 500 mL), pH 7.97; Tris-HCl, pH 7.0 (30 mM Tris); SDS-PAGE sample buffer (0.075 M Tris-HCl, pH 6.8, 10% glycerin, 2% SDS, 5% β-mercaptoethanol, 0.01% bromophenol blue).

### 2.2. Preparation of Liposomes

Liposomes (large unilamellar vesicles) were prepared as described earlier [19]. Briefly, lipid films were obtained by co-evaporation of aliquots of stock solutions of lipids in chloroform–methanol (2:1) in a round-bottom flask on a rotary evaporator, with subsequent drying for 45 min at 5 Pa. The resulting compositions (by mol) were ePC (PC); ePC–PI, 9:1 (10PI); ePC–GM_1_, 9:1 (10GM_1_); ePC–GM_1_, 9.8:0.2 (2GM_1_); ePC–CMG-PE, 9:1 (10CMG); ePC–CMG-PE, 9.8:0.2 (2CMG). If not stated otherwise, the lipid films were hydrated in phosphate buffered saline (PBS, pH 7.4), subjected to seven cycles of freezing/thawing (liquid nitrogen/+40 °C), and extruded 20 times through Whatman Nuclepore track-etched polycarbonate membranes (Cytiva, Marlborough, MA, USA) with a pore size of 100 nm on a mini-extruder (Avanti, Alabaster, AL, USA). Phospholipid concentrations in liposome dispersions were measured by the enzymatic colorimetric phosphatidylcholine assay (Sentinel Diagnostics, Milan, Italy). For anisotropy experiments, 0.025 mol% 1,3,5,7-tetramethyl-BODIPY-labeled phosphatidylcholine (TMB-PC) was added at the stage of lipid film formation.

Liposome dispersions were stored at 4 °C and used for experiments within 3 days.

### 2.3. Hydrodynamic Diameter and Zeta Potential Measurements

Particle size was measured by dynamic light scattering (DLS) with a ZetaPALS analyzer (Brookhaven Instruments Corp., Holtsville, NY, USA, helium-neon laser 633 nm, 90°). Aliquots of extruded liposomes were diluted in buffer to about 0.05 mg/mL. Samples were measured 10 times over 30 s at 20 °C. For reliable measurements of zeta potential, liposome samples with diameters of around 200 nm were prepared in 10 mM KCl, 1 mM K_2_HPO_4_, 1 mM KH_2_PO_4_, and pH 7.0 buffer (extruded 20 times through 200 nm polycarbonate membrane filters). Zeta potential values were obtained using a ZetaPALS analyzer (Brookhaven Instruments Corp., Holtsville, NY, USA; provided by the Core Facility of the Institute of Gene Biology, Russian Academy of Sciences, Moscow, Russia). Samples of the liposomes (0.16 mL, 1 mg/mL total lipids) were equilibrated for 1 min in cuvettes before 10 runs of 25 cycles per sample were performed at 25 °C. Zeta potential values were calculated using Smoluchowski approximation.

### 2.4. PAGE with Silver Staining

Pooled fractions of the liposome–protein complexes were delipidated as described in [20]. To 100 μL of the combined fractions, 400 μL of cooled methanol was added and the mixture was centrifuged for 3 min at 9000× *g* (Hamburg, Germany). To the solution, 200 μL of chloroform was added, vigorously stirred, and centrifuged for 3 min at 9000× *g*. To the mixture, 300 μL of water was added, vigorously stirred, and centrifuged for 4 min at 9000× *g*. Approximately 700 μL of the upper aqueous phase was discarded. Then, 300 μL of methanol was added to the residue and the mixture was centrifuged for 4 min at 9000× *g*. The supernatant was decanted, and the precipitate was evaporated to dryness on a rotary evaporator. The samples were dissolved in 45 μL of the reducing sample buffer, stirred, and boiled 2 × 2 min. SDS-PAGE was performed in 6% concentrating and 12% separating gels on a VE-2M Tank (Helicon, Russia) for 60 min (setup: preliminary electrophoresis, 6 min at 10 mA; concentrating, 20 min at 18 mA; separation, 34 min at 32 mA. The Pierce™ Prestained Protein MW Marker (Thermo Fisher Scientific, USA) was used as a molecular weight marker. BSA solution was used as a positive control. Proteins were visualized by silver staining. Gel was washed in bidistilled water and placed in a fixing solution (25 vol. % isopropyl alcohol, 10 vol.% acetic acid in water) overnight at +4 °C with constant stirring. Then gel was washed until it lost hydrophobicity and placed in 10% glutaraldehyde solution for 30 min at room temperature. Afterwards it was washed again with water thrice for 20 min. Staining was conducted in freshly made AgNO_3_ solution (2.5 mL NH_4_OH, 0.8 mL NaOH 10%, 0.8 g AgNO_3_ made up to 100 mL with bidistilled water). Then the gel was washed thrice for 5 min and moved to the developer solution (0.1 mL 10% citric acid and 0.936 mL 4% formaldehyde made up to 200 mL with water) for 3–5 min and then moved to a stop solution (40 % ethanol, 10% acetic acid). All procedures were carried out under constant stirring on a shaker (ELMI, Latvia).

The final image was obtained using the non-linear inversion function in the GIMP 2.10.30 software package.

### 2.5. Determination of the Protein Binding (P_B_) Values

Liposomes (20 mM total lipids; 60 μL) were incubated with 240 μL of BSA (58.5 mg/mL) at 37 °C for 15 min. The mixture was applied to a CL-4B Sepharose column (20 mL) and eluted with PBS. After the void volume of ~6 mL, fractions of 300 μL were collected. Liposome elution was monitored by absorbance at 210 and 280 nm using the NanoDrop OneC (Thermo Fisher Scientific, Waltham, MA, USA) instrument. To determine the amount of liposome-bound albumin, four fractions with the highest liposome content were pooled and concentrated to ~100 mL by ultrafiltration in Spin-X UF 500 MWCO 5000 concentrators (Corning, Oneonta, NY, USA) for 60 min at 4 °C, 8000× *g* (Eppendorf Centrifuge 5415). In the concentrated pooled fractions of the liposome–protein complexes, the amount of protein was determined using the modified Lowry procedure [21] and phospholipids using the enzymatic colorimetric phosphatidylcholine assay (Sentinel Diagnostics, Milan, Italy). Particularly, 3 μL of a fraction and 150 μL of the working enzyme solution (phospholipase D, >1500 U/L; choline oxidase, >7500 U/L; 4-aminoantipyrine, 1.2 mM; peroxidase, >7000 U/L; TES, 50 mM, pH 7.6; hydroxybenzoic acid 12 mM; EDTA, 1.3 mM; sodium azide, <0.1%) were added per well in a 96-well plate. The mixture was incubated at 37 °C for 10 min. Optical density was read at 540 nm using a Multiskan FC (Thermo Fisher Scientific, Waltham, MA, USA) microplate photometer. The amount of phosphatidylcholine in the samples was determined using the calibration curve for ePC dispersions in PBS. The P_B_ values were calculated as g protein/mol of lipids. The experiment was repeated twice with independent batches of liposomes.

BSA quantity per liposome was calculated from protein and lipid concentrations with the consideration of number of lipids per given liposome size.

### 2.6. Fluorescence Measurements

#### 2.6.1. ANS Fluorescence

Microhydrophobicity of the liposomal surface was estimated using ANS fluorescence as described in [22]. Anilino-1-naphthalenesulfonic acid ammonium salt (ANS) was dissolved in PBS, then diluted to 5 × 10^−5^ M concentration. Equal aliquots of liposomes (5 mM total lipids) and ANS (5 × 10^−5^ M) were mixed using a vortex (Biosan, Latvia) and transferred to a quartz cuvette. ANS emission spectra (λex 350 nm, emission from 400 nm to 600 nm) were obtained in a temperature-controlled cell of an F-4000 (Hitachi, Japan) fluorescence spectrometer at 25 °C. Due to liposome binding, ANS emitted signal at λmax 484 nm. Emission spectra for ANS without liposomes and vice versa were obtained under the same conditions. None of them had significant peaks. All values were obtained in triplicate.

#### 2.6.2. Anisotropy

The fluorescence anisotropy (*r*) of 1,3,5,7-tetramethyl-BODIPY-labeled phosphatidylcholine (TMB-PC) in liposomes was studied with BSA (0.5 mg/mL final concentration) and without the protein in the medium for several lipid concentrations (0.15, 0.46, 0.8, and 1.6 mM).

Liposomes with 0.025 mol %. TMB-PC were prepared. When the ratio of the probe molecules to lipids is 1/4000, no energy transfer is observed between neighboring TMB fluorophores, and the fluorescence anisotropy depends only on the fluorophore mobility [23]. Fluorescence anisotropy of TMB-PC was obtained in a temperature-controlled cell of an F-4000 (Hitachi, Japan) fluorescence spectrometer at 25 °C at λex 475 nm, λem 505 nm. All anisotropy values were obtained in triplicate.

Fluorescence anisotropy was calculated using the formula:*r* = (I_VV_ − G × I_VH_)/(I_VV_ + 2 × G × I_VH_),
where I is the intensity of the fluorescence signal at different orientations of the polarizers. The orientation of the polarizers is indicated by subscripts V (vertical) and H (horizontal). The orientation of the polarizer first is given on the excitation side and then, on the emission side. Factor G is defined as I_HV_/I_HH_.

#### 2.6.3. Calcein Release

The stability of liposomes in the presence of albumin was investigated using the dye leakage assay as previously described [16]. To prepare liposomes with calcein in a self-quenching concentration, lipid films were hydrated with 80 mM calcein solution in PBS. After extrusion, unencapsulated calcein was separated from calcein-containing liposomes using size exclusion chromatography on a column with Sephadex G-50 (1.3 cm × 15 cm) equilibrated in PBS. An aliquot of liposome dispersions (200 μL) was applied onto the column and after the void volume (~4.2 mL), fractions of 150–200 µL were collected. Fractions with the highest liposome content were pooled. Calcein concentration in combined fractions was determined by spectrophotometry (λmax 504 nm, ε 74,000 M^−1^ cm^−1^).

An aliquot of calcein-containing liposomes (5–7 μL) was diluted with pre-heated (37 °C) PBS or BSA (40 mg/mL) to a concentration of 10^−4^–10^−5^ M and incubated on a water bath at 37 °C for 0, 15, 30 min, 1, 2, and 4 h. In order to obtain initial kinetics at different BSA concentrations, 0.4, 4, or 40 mg/mL same aliquots of liposomes and protein solution in PBS were incubated in a closed quartz cuvette in a temperature-controlled cell of an F-4000 (Hitachi, Japan) fluorescence spectrometer at 37 °C for 12.5 min. The fluorescence intensity of calcein was determined before and after the liposome lysis with 30 μL of 10% Triton X-100 solution added per 300 μL of the dispersion (λex 485 nm, λem 509 nm). The fraction of calcein released before the addition of the detergent was calculated using the formula:CR = (I_t_/I_max,t_ − I_0_/I_max,0_) × I_max,t_/(I_max,t_ − I_0_) × 100%,
where I_0_ is the fluorescence intensity immediately after the liposome dilution; I_max,0_ is the fluorescence intensity upon Triton X-100 addition immediately after the liposome dilution; I_t_, fluorescence intensity of diluted liposomes after incubation for time t; I_max,t_, fluorescence intensity upon Triton X-100 addition to diluted liposomes after incubation for time t. The CR values obtained in triplicate were used to plot the relative increase in the fluorescence of calcein as function of the incubation time.

### 2.7. ATR-FTIR Spectroscopy

ATR-FTIR spectra were recorded using a Bruker Tensor 27 spectrometer equipped with a liquid nitrogen-cooled MCT (mercury cadmium telluride) detector. Samples were placed in a BioATR-II temperature-controlled cell with a ZnSe ATR (attenuated total reflection) element (Bruker, Germany). The ATR-FTIR spectrometer was purged with a constant flow of dry air. ATR-FTIR spectra were registered from 900 to 3000 cm^−1^. For each spectrum, 80 scans were accumulated at 20 kHz scanning speed and averaged. All spectra were registered in PBS, pH 7.4, at 37 °C; lipid concentration was 12 mM; volume applied onto the ZnSe element was 40 μL. To study changes in protein and lipid spectral characteristics upon incubations, spectra of pure BSA, pure liposomes, and BSA–liposome mixtures were recorded [19]. Albumin spectra were obtained for a 5 μL BSA aliquot (50 mg/mL) diluted with 35 μL PBS (pH 7.4). Intact liposome spectra were obtained for a 10 μL liposome sample aliquot (50 mM) diluted with 30 μL PBS. To study the effect of albumin on the bilayer, 5 μL BSA and 10 μL liposomes were mixed, sample volume was adjusted to 40 μL with PBS, and afterwards the mixture was incubated on the ZnSe element. Spectra of liposome–protein complexes were recorded after 3, 5, 10, 15, and 20 min of incubation. The experiment was repeated twice with independent batches of liposomes. Spectral data were processed using the Opus 7.5 (Bruker, Germany) software system, which includes linear blank subtraction, straight-line baseline correction, and atmosphere compensation. If necessary, Savitsky–Golay smoothing was used to remove white noise and wide peaks were subjected to a deconvolution procedure. Peaks were identified by the standard Bruker peak picking procedure.

### 2.8. Statistical Analysis of Experimental Data

All experimental data were analyzed using QtiPlot 0.9.8.9. svn 2288 software by Ion Vasilief. Data obtained by fluorescence measurements (e.g., calcein release at different time points) were expressed as mean ± standard deviation (SD) to show dispersion of the experimental values. Data obtained from FTIR spectra and P_B_ evaluation were expressed as mean ± standard error (SE) to control precision of the estimate when we compared sample characteristics. To evaluate how albumin structure changes during incubation with liposomes, we used a two-sample *t*-test with the following parameters: independent test; significance level 0.05; confidence intervals at 95%. The null hypothesis was that BSA structure during incubation with liposomes is the same as at 0 min without them. The alternative hypothesis was that they differ from each other.

## 3. Results and Discussion

### 3.1. Liposome Preparation and Characteristics

Liposomes were prepared with egg phosphatidylcholine as a main lipid and phosphatidylinositol (PI), ganglioside (GM_1_), or carboxymethylated oligoglycine (CMG) lipid conjugate (CMG-PE) as additives (2 or 10 %). The structures are presented in Figure 1.

Measured size and zeta-potential values of liposomes under investigation are presented in Table 1. All samples were negatively charged with ζ-potential ranging from −12 mV for PC as the least charged sample to −62 mV for liposomes with 10% CMG-conjugate.

### 3.2. Albumin Adsorption Evaluation

To determine the affinity of albumin to various bilayer compositions—is it strongly bound or is it readily exchangeable—liposomes were incubated in a solution with average albumin physiological concentration (~46 mg/mL). Then, we separated liposome–protein complexes from bulk protein using size exclusion chromatography (SEC). This isolation technique affects weakly bound proteins as they can detach from the complex.

At first, protein binding was visualized by gel electrophoresis with silver staining (Figure 2). All liposome samples had tightly bound albumin on their surface.

The protein binding (P_B_) value was calculated as a relation between protein mass (g) in liposome–protein complexes and lipid molar quantity (mol) in said complexes (Table 1). This value is often used for liposome–protein complexes’ characterization and was introduced by Semple et al. [24] as a way to evaluate the rate of liposome clearance. The lower the P_B_ value, the slower is liposome clearance. After incubation with albumin, all liposomes had low yet quite similar P_B_ values. Some decrease could be seen for 10GM_1_ liposomes when compared to PC. Ohtsuka et al. in [25] also observed a decrease in BSA binding to dipalmitoylphosphatidylglycerol (PG) liposomes with 10% GM_1_ compared to a pure PG sample, presumably due to the increase in hydrophilic properties and surface coverage with sugar chains. Thus, the effect is unlikely due to only electrostatic interactions (oligosaccharide moiety of GM_1_ has a negatively charged sialic acid fragment).

To better understand the degree of albumin adsorption, we calculated the number of bound albumin molecules per single liposome (Table 1). 

The theoretical capacity of the liposome surface for albumin was calculated based on the fact that an average 110 nm phosphatidylcholine liposome should contain about 97,700 lipid molecules and its surface area should be around 38 000 nm^2^. As for albumin, it is considered to be a 14 × 4 × 4 nm^3^ ellipsoid of revolution [26]. Thus, if albumin is adsorbed onto liposomes as an ellipse with 14 and 4 nm axes, then ~216 molecules can fit the lipid surface. If it is bound as a 4 nm × 4 nm ellipse, then theoretically ~756 molecules could fit the liposome. Such a dense albumin adsorption is scarcely probable. For example, liposome–protein complexes isolated after liposome incubation in plasma are believed to be multilayered and contain a maximum of 200–300 protein names, with each protein type represented by a few molecules [4]. In our experiment, liposomes bound very few albumin molecules, i.e., from four to nine (Table 1). Like with P_B_ values, liposomes containing 10 mol.% of the GM_1_ ganglioside (four molecules) differed from PC (eight molecules) the most. All in all, these quantities are scarce in comparison with our theoretically calculated range. Interestingly, when Kristensen et al. [27] incubated PEG-containing liposomes in 1 mg/mL solution of human serum albumin, they detected protein saturation levels being two to five molecules per liposome.

### 3.3. Fluorescence Anisotropy in Different Bilayers

The fluorescence anisotropy (*r*) reflects fluorophore mobility in a given membrane and is changing from zero, when the fluorophore rotates freely, to 0.4 when it is fixed. We determined fluorescence anisotropy of a phospholipid probe TMB-PC in our liposomes before and after albumin addition. All samples have fluid-phase bilayer with *r* value about 0.13–0.14 except for the 10GM_1_ sample (Figure 3a). For these liposomes, the initial *r* value was lower, about 0.10. This could be due to the fact that the increase in ganglioside percentage causes an increase in mean cross-sectional area per lipid value in the membrane [28] which therefore decreases mean chain density as GM_1_ molecules need more headspace than PC in membranes. As a result, TMB-PC rotates more freely in the bilayer with 10% GM_1_ leading to a decrease in the anisotropy signal. The sample with 2% GM_1_ did not differ from PC. In [28], the authors stated that lower content of GM_1_ (less than 5%) has a rather elusive effect on the palmitoyloleoylphosphatidylcholine (POPC) membrane. For example, it does not affect POPC phase transition temperature, although it slightly decreases bending rigidity of POPC membranes.

Albumin adsorption scarcely affected viscosity of all liposome membranes. Due to albumin adsorption, anisotropy values decreased by 5–10% independently of the increase in protein concentration (Figure 3a). These data can be interpreted in two ways. On one hand, if liposome incubation with albumin at two protein:liposome ratios, namely, 420 albumin molecules per liposome (Figure 3a, left) and 4550 albumin molecules per liposome (Figure 3a, right), results in the same impact on the membrane, it could be an indication that the amount of protein bound on the surface in both cases is similar. Moreover, this amount should be equal to 420 BSA molecules per liposome or lower. Given that we detected less than 10 tightly bound protein molecules (Table 1), it is possible that this saturation level lies between 420 and 10 protein molecules per liposome but cannot be specified further with our current methods. On the other hand, since fluorophore position gravitates toward hydrophobic region with lower lateral pressure [29], scarce anisotropy changes could result from membrane hydrophobic area insensitivity to weakly bound proteins. Thus, if albumin molecules do not affect the order of hydrophobic chains, we are not able to detect changes upon protein adsorption using anisotropy measurements.

### 3.4. Microhydrophobicity of Liposomal Surface Estimated by ANS Fluorescence

Albumin binding was considered to depend on hydrophobic interactions in the work of Kristensen et al. [27] as it has high affinity for binding hydrophobic membrane packing defects in PEGylated gel-phase DSPC liposomes [27]. Also in the work of Yokouchi et al. [22], albumin chooses to bind more hydrophobic bilayer among equally charged liposomes. In this work, we decided to evaluate the microphobicity of our liposomes using the same technique with ANS [22]. Anilino-1-naphthalenesulfonic acid (ANS) fluorescence properties in water differ from those in hydrophobic media. The ANS fluorescence signal in buffer solution was almost indistinguishable from background noise (less than 2 a.u., data not shown). However, when ANS mobility is restricted and it is bound by hydrophobic interactions it results in spectral signature changes, namely, the blue shift occurs, fluorescence intensity amplifies, and lifetime increases [30]. In our experiment (Figure 3b) the ANS signal was almost identical upon binding to PC, 10GM_1_, or 2CMG samples. The addition of 2 mol. % GM_1_ to PC caused a moderate increase in the signal and 10 mol. % PI or CMG-PE, on the contrary, caused a significant reduction in the ANS signal and its binding.

ANS fluorescence may decrease if less hydrophobic surface area is available for binding or due to electrostatic repulsion between ANS and the negatively charged surface [22].

If all our samples had approximately the same hydrophobic surface area and our additives affected its accessibility by electrostatic repulsion only, then the ANS signal should have been a ζ-potential function with linear trend in Figure 3c, and this is not the case. 

In our batch, the PC sample had closest to zero ζ-potential value and should have created minimal repulsion for the fluorophores. Ganglioside GM_1_ shifted ζ-potential values towards higher electronegativity but it did not affect ANS adsorption as expected. Supposedly ganglioside increases microhydrophobicity due to the same mechanism which affected anisotropy. By increasing the ganglioside ratio in the bilayer, we decreased mean chain density and helped to create new hydrophobic spots [28] that attract ANS and increase its fluorescence signal. Concurrently, GM_1_ also increased net charge of the liposomes, creating repulsion between the surface and ANS molecules. As a result, we observed an increase in the fluorescence signal for 2GM_1_, where the shifted ζ-potential value is still close to the PC sample and the same level of fluorescence signal for 10GM_1_, with three-fold lower ζ-potential (Figure 3c) compared to PC. 

Liposomes with 2% CMG-PE showed the same level of fluorescence anisotropy intensity and ANS fluorescence intensity as the PC sample (Figure 3a,c), although the ζ-potential value of the former is equivalent to that of the 10GM_1_ liposomes (both –38 mV). We believe that carboxymethyl groups prevent oligoglycine from secondary structure formation or mushroom-like conformation of the polar moiety due to steric hindrance and electrostatic repulsion; thus, CMG is forced to stay in an antenna-like conformation. This yields rather sparse coverage of the lipid membrane, while ζ-potential values remain significant. Therefore, ANS molecules could pass between the CMG moieties and bind to hydrophobic spots on the surface of 2CMG liposomes as effectively as on the PC ones. By increasing CMG-lipid content up to 10% we concurrently increased lipid membrane coverage and doubled the ζ-potential, thus amplifying electrostatic repulsion between ANS and the bilayer surface and narrowing the space for ANS passage between stretched out carboxymethylated peptide chains. After all, the probe can still reach some hydrophobic spots on the 10CMG surface as the fluorescence signal was higher for that sample than for 10PI (Figure 3c). Liposomes with phosphatidylinositol had intermediate ζ-potential values (Table 1) yet exerted the highest repulsion level for ANS. The PI molecule does not have a bulky polar head group similar to ganglioside or protruding polar chains similar to CMG; all of its charged groups are closer to the membrane, thus preventing the probe adsorption on hydrophobic areas.

### 3.5. Assessment of the Liposome Stability during Protein Adsorption

In our previous work with PI, CMG-conjugate, and GM_1_ [16], we observed that in most cases protein adsorption could still destabilize the lipid membrane. As our work was conducted in plasma, we were not able to simultaneously differentiate between various proteins effects that can trigger dye release from the liposome’s inner volume. According to published works, there are several protein candidates. For example, in [31], the authors reported that IgM binding to PEG-containing liposomes followed by complement activation can cause dye release from the inner liposome volume. Other researchers reported that adsorbed albumin can penetrate the lipid bilayer [32] and enhance liposome cargo release [33].

Here we used a calcein release assay to study stability of lipid bilayers and leakage of solutes from the interior of liposomes during albumin adsorption. We prepared samples of liposomes listed in Table 1 with calcein encapsulated at self-quenching concentration and incubated them in PBS or 40 mg/mL albumin solution at 37 °C. Dilution with buffer caused moderate leakage of dye through fluid membranes (less than 5% CR in 15 min), which did not affect further stability of all samples except 10CMG (Figure 4a). For this liposome sample, the amount of free calcein doubled during the following 4 h of incubation in buffer. When liposomes were injected into albumin solution, 10CMG was the only sample that quickly discharged a substantial amount of calcein (Figure 4b).

We believe that calcein leakage is not equal to liposome destruction in this case, as earlier we observed almost complete dye loss from CMG-containing liposomes in plasma while we could still detect Forster resonance energy transfer between two BODIPY-based probes in the bilayer [16]. In 10CMG, electrostatic repulsion between neighboring CMG molecules and CMG–albumin interactions upon protein adsorption create lipid oscillations resulting in transient pore formation in the bilayer. This facilitates the dye transfer across the bilayer. For the 2CMG sample with scattered oligopeptide-conjugates and lower ζ-potential value, an increase in the calcein fluorescence signal was also observed, yet it only slightly surpassed 5% and was similar to that of the 10PI sample. 

As liposomes with the 10% oligopeptide conjugate released a sufficient percentage of calcein minutes after albumin addition, we decided to look into initial kinetics of this process for several albumin concentrations (Figure 5).

As a control, we used PC liposomes, which sustained interactions with albumin (Figure 5, red lines). The most diluted albumin concentration was 0.4 mg/mL, which gave us the same order of protein/liposome ratio of about 10 molecules per liposome. Calcein release from 10CMG decelerated after 5 min in diluted media and the final value of CR was similar to that in buffer (Figure 4a and Figure 5a). A 10-fold increase in the protein concentration (4 mg/mL) affected liposome stability and caused acceleration of dye leakage (Figure 5b) similar to that at a physiological concentration of the protein, 40 mg/mL (Figure 5c). These two concentrations caused active calcein release, resulting in sharp growth of the fluorescence intensity in Figure 5b,c moments after addition to protein. We believe that similarities in the behavior between liposome–protein complexes formed in 4 and 40 mg/mL albumin solutions in terms of the dye release can be an indirect indication of the protein saturation level. Then, supposedly the surface of 10CMG is saturated with 10 to 120 molecules of albumin per liposome.

As albumin has a drastic effect on 10CMG stability, it should have been at least one of the proteins that caused full dye leakage from liposomes containing 10% CMG-conjugate in our previous experiments [16].

### 3.6. ATR-FTIR Spectroscopy

All methods mentioned above provide information about the lipid membrane but not the protein. In order to follow the state of the protein as well as the lipid membrane during incubation, we used ATR-FTIR spectroscopy. First, ATR-FTIR spectra were recorded for each liposome suspension in PBS buffer. Then we incubated the liposome–albumin mixtures in the cell and recorded spectra each 5 min for 20 min to follow time- delayed changes in the protein and the bilayer. The spectra were also recorded for albumin in buffer as a reference for protein structure at different time points.

All changes occurred within the first 5 min of incubation; afterwards there were no significant differences between the obtained spectra. Furthermore, there were no significant differences in albumin structure when incubated without liposomes. Protein structure was calculated from the Amide I band (1620–1680 cm^−1^) analysis (Figure 6a). To track changes in the lipid bilayer we analyzed the following peaks in the ATR-FTIR spectra: asymmetric stretching vibrations of the phosphate (1220–1260 cm^−1^) and carbonyl groups (1720–1750 cm^−1^), and asymmetric (~2920 cm^−1^) and symmetric (~2850 cm^−1^) stretching vibrations of the CH_2_ groups (Figure 6b). Unfortunately, we were unable to track changes in ganglioside, CMG-conjugate, and phosphatidylinositol moieties as they have low intensity, at the level of background noise.

#### 3.6.1. Changes in the Structure of the Lipid Bilayer

*Phosphate groups*. Lipid phosphate group oscillations are presented by several signals (Figure 6b): the PO_2_^−^ group has asymmetric stretching vibrations (1220–1270 cm^−1^) and symmetric stretching vibrations (split peak at ~1090 and 1065 cm^−1^), and P–O stretching vibrations give a small peak of ~820 cm^−1^ [34]. According to [35], the phosphate group is the main hydration site for lipids in the polar region and is sensitive to inter- and intramolecular binding. In [35], the hydrogen bonds between water molecules and the PO_2_^−^ group were disrupted by anesthetics addition. The latter displaced the hydrate shell and bound instead of it, which resulted in the asymmetric stretching vibrations peak shift to the higher wavenumber region. If the phosphate group is fully dehydrated, then its peak shifts to 1260 cm^−1^ [35]. 

We analyzed changes in the peak position of the PO_2_^−^ group asymmetric stretching vibrations after albumin addition. In less than 5 min, this peak shifted to the higher wavenumber region, which means that PO_2_––H_2_O bonds were replaced by PO_2_––BSA bonds to some extent. Table 2 shows the average peak shift after BSA addition to the liposomes. Albumin had the smallest effect on the polar region of the pure PC sample. In ganglioside liposome spectra, the PO_2_^−^ shift was farthest regardless of GM_1_ amount (2 or 10%) in the membrane.

Although all liposomes established new bonds with the protein, it did not eliminate all water molecules from the membrane surface as new peak positions were still far from 1260 cm^−1^, suggesting that the protein did not cover the entire membrane. However, as there were no significant changes in the peak shape for all the samples, either only few groups interact with the protein and we cannot detect them as a new population or loosely bound albumin, which we were not able to detect after SEC, is uniformly distributed across the membrane and thus all phosphate groups have a protein molecule in their vicinity (Appendix A).

*Choline groups*. The choline group peak is clearly visible in the spectra of liposomes (~970 cm^−1^, Figure 6b). According to [35], this group can respond to changes in the hydrate shell as well, but the signal is much weaker than that from the phosphate group. A slight shift in the choline peak position (~1 cm^−1^) after albumin addition was observed only for liposomes with 10% phosphatidylinositol (Figure 7).

*Ester carbonyl stretching band*. Carbonyl function C=O is a part of the glycerol core, which is a deeper part of the lipid membrane than the phosphocholine region. Nonetheless, C=O groups can also bind some water molecules [35]. The 1720–1750 cm^−1^ band presents carbonyl group stretching in FTIR spectra (Figure 6b). The fully hydrated lipid bilayer has a signal ~1730 cm^−1^, which is a superposition of two components: *sn-1* and *sn-2* C=O. Upon deconvolution, *sn-1* components show a peak at ~1742 cm^−1^; they are believed to be free of hydrogen bonds and orientated towards the inner part of the bilayer. Meanwhile, *sn-2* carbonyls are highly hydrated and orientated towards the polar region; their peak position after deconvolution is ~1727 cm^−1^ [36,37,38,39]. The peak at ~1738 cm^−1^ represents *sn-2* C=O free of hydrogen bonds [36]. When lipids bind other molecules and loose water, this peak becomes more substantial [40]. In the work of Cieślik-Boczula et al. [41], fully hydrated liposomes had a signal at ~1725 cm^−1^ in the spectrum, and for dry films it shifted to 1745 cm^−1^.

In order to analyze the signals of carbonyl groups, the spectrum in the region 1700–1760 cm^−1^ (Figure 8a) was deconvoluted and the percentage of hydrogen-bound carbonyl groups was calculated from peak areas (Figure 8b). After deconvolution, the main subcomponents were positioned at ~1731, ~1738, and ~1742 cm^−1^ (Figure 8a), which is in accordance with previously mentioned works [36,37]. Some minor subcomponents were also obtained during deconvolution. Interestingly, free *sn-2* C=O groups (~1738 cm^−1^) were detected even before protein addition.

Albumin binding to the liposome surface did not cause significant changes in hydrated carbonyl groups’ percentage; however, during incubation we observed changes in the peak shape—its splitting into three subcomponents mentioned earlier and an additional shoulder at ~1714 cm^−1^ (Figure 8c, full description of the deconvolution components is available at the Zenodo repository [42]). According to [43] and [44], this shoulder belongs to a carbonyl group participating in two hydrogen bonds. The most pronounced changes in the C=O part of the spectrum were observed for 10PI liposomes. This splitting shows that due to protein adsorption, lipid carbonyl groups divide into several populations depending on the quantity of hydrogen bonds with protein and with water.

*Methylene stretching vibrations*. Methylene stretching vibrations (2800–3000 cm^−1^) are generally the strongest bands in the spectra of lipids (Figure 6b). According to [34], methylene group stretching bands are not affected by oscillation changes in the polar region and are less prone to peak overlapping.

Methylene asymmetric and symmetric stretching vibration frequencies are sensitive to the *trans/gauche* orientation changes in the lipid acyl chains. During lipid phase transition from the gel to the liquid ordered phase, the number of *gauche* conformers in the chain increases, causing higher disorder in the chain region. Thus, the frequency of stretching vibrations of methylene groups increases. This process leads to a shift in the peaks towards higher wavenumbers. The absorption band of asymmetric stretching vibrations shifts from ~2916 to ~2924 cm^−1^ [45,46]. The frequency of asymmetric CH_2_ stretching vibrations in our liposomes was ~2924 cm^−1^ in accordance with the data for the PC liquid disordered phase [46]. Incubation with albumin did not affect methylene vibrations in any liposomes (Appendix A) except for 10CMG. For this sample, we observed a shift in asymmetric vibrations’ peak position of ~1 cm^−1^ towards a more disordered state (Figure 9). Probably, these very changes in the 10CMG bilayer upon protein adsorption, although small, could facilitate calcein release from the liposomes.

Taking into consideration all the spectroscopy data for the liposomes, we can conclude that albumin adsorption affects mostly polar headgroups disrupting hydrogen bonds with water and creating new ones with the protein instead. As we did not observe significant dehydration of ester carbonyls, we have no reason to believe that albumin penetrates to the glycerol core upon its binding let alone further to the acyl chains region.

#### 3.6.2. Changes in the Protein Structure

As was mentioned before, albumin structure was calculated from the Amide I peak in the FTIR spectra. Similar to methylene peaks in lipids, Amide I and Amide II are the most prominent peaks in the protein spectra sensitive to the structural dynamics of proteins [47]. The Amide I mode (1600–1700 cm^−1^) almost entirely consists of C=O stretching vibrations of peptide bonds, while Amide II represents in-plane NH bending and CN stretching vibrations. Different hydrogen bonding and C=O geometry in α-helix, β-sheet, β-turn, and random coil give rise to different components in Amide I. Secondary structure is calculated from these components upon the assumption that the protein can be considered as a linear sum of its structural elements and the percentage of these elements is related only to spectral intensity [48]. Changes in the secondary structure of the protein, when it interacts with the membrane, can be detected as a redistribution of the signal intensities of different structural elements, such as α-helix/β-sheet/β-turn/random coil, in the Amide I mode (in our case, 1628–1685 cm^−1^) [19]. The appearance of protein aggregates in the solution corresponds to the absorption peaks of the antiparallel β-sheet in the frequency regions of approximately 1624 and 1696 cm^−1^ [49]. These individual signals of structural elements are not distinctive in spectra and they are obtained after second derivative deconvolution of the Amide I mode. In our work, a correlation between secondary structure and its subcomponent peak positions was made mainly based on [49,50,51]. Assignment of Amide I deconvolution components to corresponding structure elements was made as shown in Table 3.

If any conformational changes occurred in the protein to adapt it to binding the lipid membrane, they were happening very rapidly. Starting from the 10th minute of incubation with liposomes, we did not observe any crucial changes in the protein structure. Supposedly, after albumin has already found its equilibrium conformation and bound to the surface, no further changes occurred. If so, all insignificant fluctuations in data after 10 min are a result of measurement and deconvolution errors. Calculated percentages of each of the secondary structure elements (α-helix/β-sheet/β-turn/random coil) for albumin before and after interaction with the liposomes are presented in Table 4. Amide I region spectra for different liposomes are presented in Appendix A.

According to the calculations, the distribution of structural elements in a control sample of albumin without liposomes at the beginning of incubation (0 min) and after 10 min at 37 °C differs by a couple of percent, and *p*-values exceed the threshold of 0.05 (Table 4), so we consider the differences insignificant. The structure of albumin bound to the surface of PC, 2CMG, and both GM_1_-containing liposomes was also similar to the original free protein and its 10 min control. 

The most pronounced and significant changes in albumin structure occurred during incubation with a sample containing 10% phosphatidylinositol. Upon binding to the surface, albumin lost 7% of the α-helix and a couple % of β-components. At the same time, the proportion of random coil increased by ~11%, which indicates partial denaturation of the protein. A similar effect was observed for a sample with 10% CMG-conjugate. The redistribution of intensity from the α-helix peak to the random coil peak during denaturation was observed by the authors [52] for horseradish peroxidase. This is an undesirable consequence of albumin binding because, as was mentioned before, it can cause protein recognition by scavenger receptors [7]. More significant changes in protein structure were observed in our previous work for positively charged liposomes [19].

Interestingly, the degree of albumin denaturation on the liposome surface corresponds to microhydrophobicity measurement results. When adsorbed on the most hydrophobic samples (PC, 2GM_1_, 2CMG, and 10GM_1_), BSA has the ability to preserve its structure. Whereas when it is absorbed onto the most polar sample (10PI), we see the highest conversion of α-helices to random coil. Possibly, due to denaturation, albumin creates different bonds with the bilayer, and this is the reason why we detected a slight shift in the choline group peak only for 10PI, although this denaturation can be evaded by electrostatic coupling of PI to positively charged molecules [19].

## 4. Conclusions

Although the amount of albumin molecules tightly bound to the liposomes was rather low—less than 10 BSA molecules per liposome—these proteins could still affect membrane stability and vice versa. The most stable samples were the ones with characteristics similar to those of the pure phosphatidylcholine membrane by microhydrophobicity evaluation. Semi-rigid carboxymethylated oligoglycine lipid conjugate (10 mol. %) was not beneficial neither for liposome membrane stability nor for protein binding neutralization. It caused significant permeability of the lipid bilayer even at an albumin concentration 10-fold lower than physiological and disrupted protein folding. Both these consequences of protein adsorption would make the drug delivery system ineffective. Considering liposome behavior in all our experiments, only ganglioside GM_1_ showed desirable features for liposome shielding from protein as it lowered albumin adsorption at 10 mol. % and did not affect the protein structure during incubation.

## Figures and Tables

**Figure 1 membranes-12-01031-f001:**
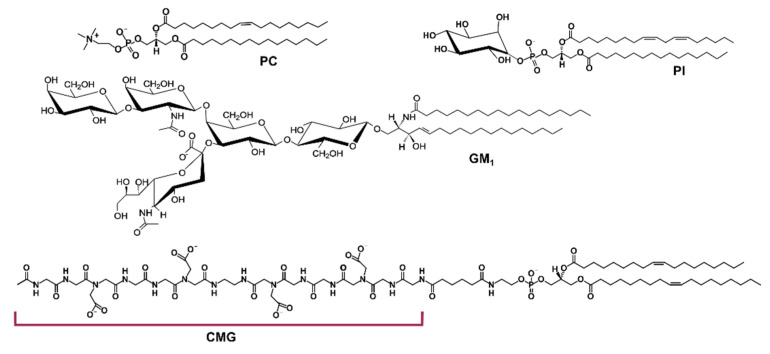
Chemical structures of the components of liposomes used in this study. Representative structures of egg phosphatidylcholine (PC), soybean phosphatidylinositol (PI), ganglioside GM_1_ from bovine brain and carboxymethylated oligoglycine (CMG) lipid conjugate are shown.

**Figure 2 membranes-12-01031-f002:**
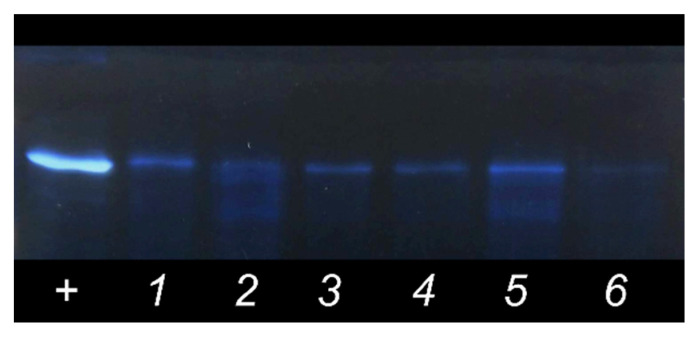
Protein visualization by gel silver staining with color inversion. Lanes are as follows: BSA solution 0.05 mg/mL (+); PC liposomes (1); 10PI liposomes (2); 2GM_1_ liposomes (3); 10GM_1_ liposomes (4); 10CMG liposomes (5); 2CMG liposomes (6).

**Figure 3 membranes-12-01031-f003:**
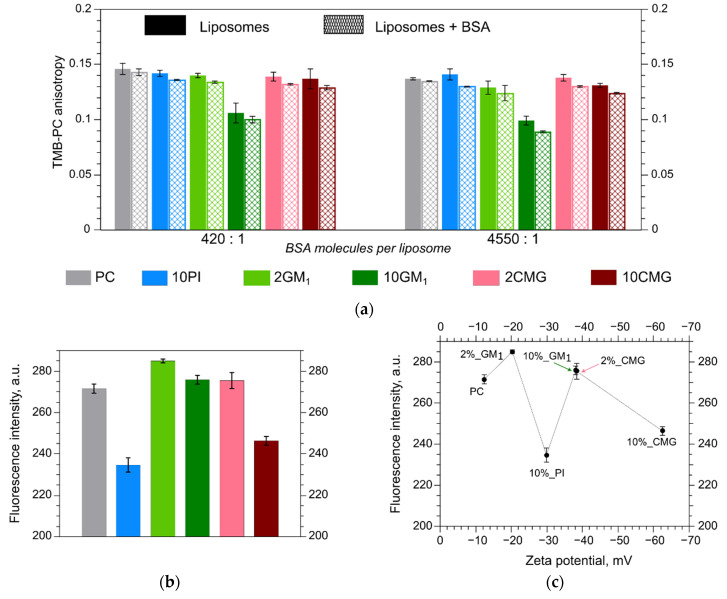
(**a**) TMB-PC anisotropy in liposomes alone and in the presence of albumin; (**b**) ANS fluorescence in the presence of liposomes; (**c**) ANS fluorescence intensity–liposome zeta potential dependence. Mean ± SD (*n* = 3) are presented.

**Figure 4 membranes-12-01031-f004:**
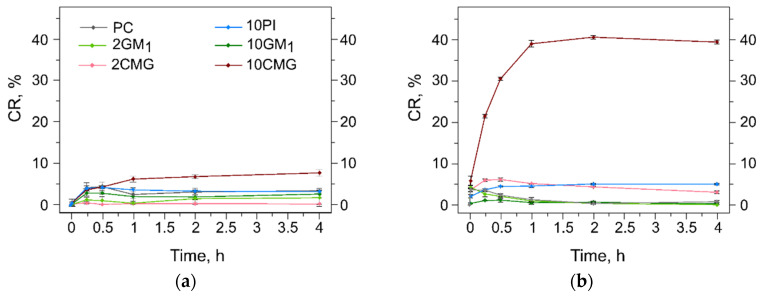
(**a**) Calcein release (CR, %) from liposomes upon incubation in PBS at 37 °C in the absence of albumin; (**b**) calcein release (CR, %) from liposomes upon incubation in PBS at 37 °C in the presence of BSA 40 mg/mL. Means ± SD (n = 3) are presented.

**Figure 5 membranes-12-01031-f005:**
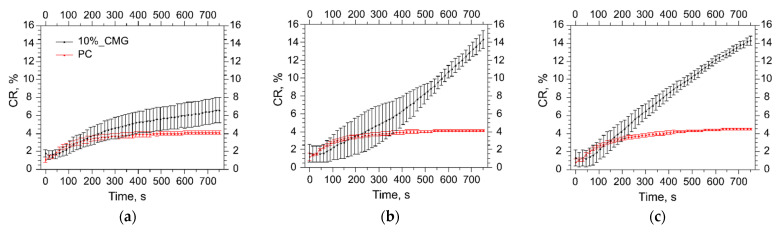
Calcein release (CR, %) from PC (red line) and 10CMG (black line) liposomes in albumin solution: 0.4 mg/mL (**a**); 4 mg/mL (**b**); 40 mg/mL (**c**). Means ± SD (n = 3) values are presented.

**Figure 6 membranes-12-01031-f006:**
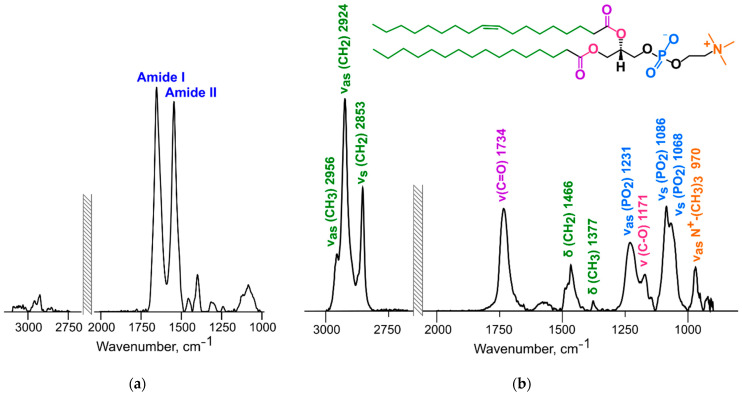
Albumin (**a**) and liposome (**b**) FTIR spectra recorded in PBS at 37 °C.

**Figure 7 membranes-12-01031-f007:**
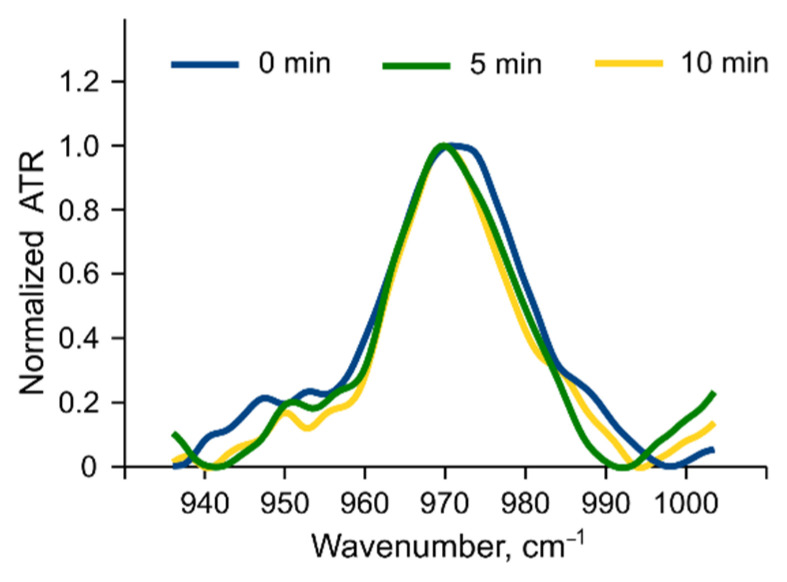
Normalized choline group peak in 10PI liposomes before (blue) and after albumin addition (green and yellow). FTIR spectra were recorded in PBS at 37 °C, lipid concentration of 12 mM, and albumin concentration of 6 mg/mL.

**Figure 8 membranes-12-01031-f008:**
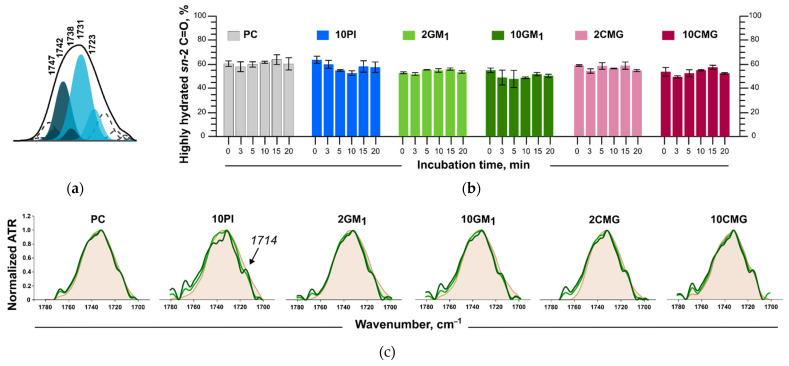
(**a**) C=O deconvolution example; (**b**) changes in the percentage of *sn-2* C=O hydrated groups during incubation with BSA; (**c**) C=O region in FTIR spectra for liposomes alone (beige) and with albumin (shades of green). FTIR spectra were recorded in PBS at 37 °C, lipid concentration of 12 mM, and albumin concentration of 6 mg/mL.

**Figure 9 membranes-12-01031-f009:**
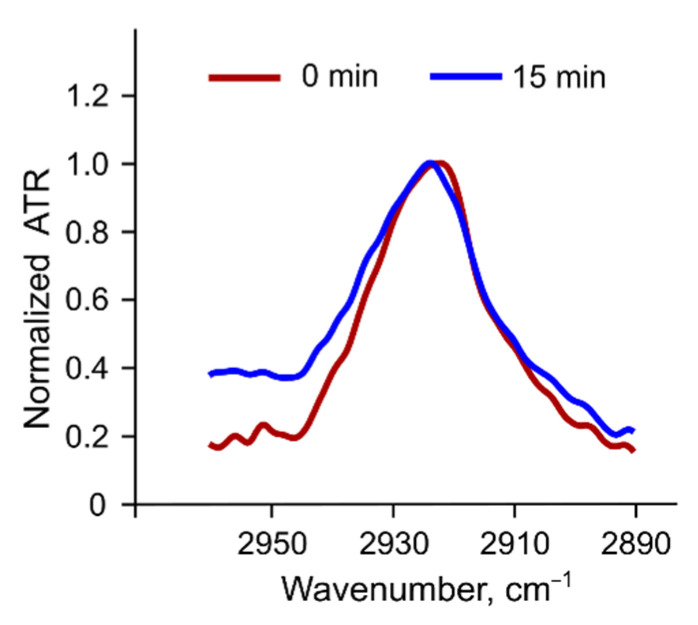
Normalized asymmetric stretching vibrations peak in the 10CMG liposomes before (red) and after albumin addition (blue). FTIR spectra were recorded in PBS at 37 °C, lipid concentration of 12 mM, and albumin concentration of 6 mg/mL.

**Table 1 membranes-12-01031-t001:** Liposome characteristics.

Liposome Sample ^1^	Mean Diameter, nm (±SD)	Mean PDI (±SD)	Mean ζ-Potential ^2^, mV (±SD)	P_B_, g_BSA_/mol_lipids_ (±SE)	BSA Molecules per Liposome (±SE)
PC	121.3 ± 2.4	0.084 ± 0.003	–12.37 ± 0.66	4.72 ± 0.60	8.31 ± 1.06
10PI	104.4 ± 1.8	0.068 ± 0.016	–29.94 ± 1.58	5.88 ± 0.71	7.84 ± 0.94
2GM_1_	113.1 ± 0.5	0.030 ± 0.012	–20.28 ± 0.59	5.20 ± 1.05	7.64 ± 1.54
10GM_1_	118.5 ± 1.3	0.092 ± 0.008	–38.13 ± 1.02	2.48 ± 0.15	4.00 ± 0.25
2CMG	107.4 ± 0.8	0.014± 0.007	–38.38 ± 1.15	3.79 ± 0.47	5.57 ± 0.70
10CMG	109.6 ± 0.7	0.099 ± 0.024	–62.54 ± 1.46	6.61 ± 0.94	9.72 ± 1.38

^1^ For designations see Materials and Methods. ^2^ As assessed with ZetaPALS analyzer (Brookhaven Instruments Corp., Holtsville, NY, USA) for 200 nm liposomes.

**Table 2 membranes-12-01031-t002:** Average PO_2_^−^ group peak positions before and after albumin addition.

Liposome Sample	Initial Peak Position in Liposomes, cm^−1^ (±SE)	Peak Position in Liposome–Protein Complex, cm^−1^ (±SE)	Average Shift, cm^−1^
PC	1230.7 ± 0.3	1231.5 ± 0.3	<1
10PI	1230.2 ± 0.1	1233.6 ± 0.4	~3.4
2GM_1_	1231.6 ± 0.2	1236.0 ± 0.2	~4.4
10GM_1_	1229.9 ± 0.5	1234.4 ± 0.6	~4.5
2CMG	1230.1 ± 0.2	1233.8 ± 0.2	~3.7
10CMG	1228.9 ± 0.5	1231.1 ± 0.2	~2.2

**Table 3 membranes-12-01031-t003:** Correlation between secondary structure element in BSA and Amide I second derivative position ^1^.

Secondary Structure Element	Peak Range, cm^−1^
α-helix	1650–1660
β-sheet	1628–1639
β-turn	1664–1687
random coil	1640–1649
Intermolecular β-sheet (aggregates)	1618–1626; 1688–1696

^1^ As previously published [19], expanded ranges for α-helix and random coil peak positions were used as proposed in [50] for albumin.

**Table 4 membranes-12-01031-t004:** Secondary structure elements calculated from FTIR spectra of BSA alone and after incubation with the liposomes. FTIR spectra were recorded in PBS at 37 °C, lipid concentration of 12 mM, and albumin concentration of 6 mg/mL. Fitting procedure was conducted using Levenberg-Marquardt method with peak type parameter as Gauss or if unavoidable as Gauss and Lorentz combination. Null hypothesis is that BSA structure during incubation is the same as at 0 min.

Sample	α-Helix %, (±SE)	*p*-Value	β-Sheet + Turn %, (±SE)	*p*-Value	Random Coil %, (±SE)	*p*-Value
BSA, 0 min	59.9 ± 1.2	–	23.2 ± 2.5	–	16.9 ± 2.0	–
BSA, 10 min	57.6 ± 0.8	0.1262	25.5 ± 0.7	0.2730	16.9 ± 0.8	0.9814
+PC	61.0 ± 0.8	0.4570	20.9 ± 1.2	0.4179	18.0 ± 1.1	0.6174
** *+10PI* **	** *52.8 ± 1.5* **	** *0.0058* **	** *19.1 ± 2.1* **	** *0.2335* **	** *28.0 ± 1.2* **	** *0.0009* **
+2GM_1_	57.1 ± 0.9	0.0872	22.9 ± 0.5	0.9128	20.1 ± 0.8	0.1562
+10GM_1_	56.9 ± 0.8	0.0664	23.5 ± 1.6	0.9069	19.6 ± 1.2	0.2657
+2CMG	58.3 ± 1.2	0.3669	24.1 ± 1.7	0.7528	18.8 ± 0.9	0.3923
** *+10CMG* **	** *55.6 ± 1.1* **	** *0.0284* **	** *18.8 ± 1.2* **	** *0.1242* **	** *26.2 ± 1.5* **	** *0.0049* **

## Data Availability

The deconvolution data presented in Figure 8 are openly available in Zenodo at https://doi.org/10.5281/zenodo.7188568, reference number [42].

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
