# Peer review of "Spectroscopy Study of Albumin Interaction with Negatively Charged Liposome Membranes: Mutual Structural Effects of the Protein and the Bilayers"

_membranes, 2022, doi:10.3390/membranes12111031_

Round 1

Reviewer 1 Report

Overall, authors have presented a great work.  This manuscript is well written and experiments were well designed with sufficient controls.  Authors put in significant effort to outline how to analyze these experimental results, very helpful to non-expert readers.  Authors presented their studies in a logical, sequential order and make it easy for readers to finish in 1/2 day.  Most importantly, authors have presented a work that describes many views on this biological event from lipid/lipid membrane perspectives, which is fully in line with what I have proposed for this special issue.

These are my questions to authors.

1.  In line 91, is there a reason why you would need to have 2 different mol/mol ratios?  Please explain further.  Thank you.

2.  In line 93, did you observe precipitation effect using PBS instead of phosphate buffer or pure water?  Normally, lipid or liposome suspension precipitate easily in PBS (and sometimes in phosphate buffer).  How did you resolve this problem?  Please explain more.  Thank you.

3.  In line 158, is it common to use OD540 to read?  Please explain what OD540 signal should indicate and what this signal change mean to help non-expert readers understand better.  Thank you.

4.  In line 186, will physiological temperature (e.g., 37oC) make differences in anisotropy in TMG-PC liposomes?  And other experiments?  25oC is good for biophysical studies, but may not be sufficient to pass phase transition temperatures (Tm) of liposomes of different composition presented in your work.  Do you know Tm of your liposomes?  Also, many interactions may occur differently or not occur at all at physiological temperature.  Please explain how you can account your current observation on this aspect.  I would recommend to run a set of experiment at 37oC to show significant/insignificant differences to conclude if these results are relevant to physiological process or mainly only to pure biophysical studies.  Thank you.

5.  In line 228, ATR-FTIR spectroscopy experiments were run at physiologically relevant temperature (37oC), which is very good, but might not be consistent with rest of studies at 25oC.  I would recommend authors to reconcile this discrepancy, but this does not affect my recommendation to publish this paper.  Thank you.

6.  In line 295 ~ 300, you stated that low number of albumin molecules bound to liposomes may be due to steric hinderance, but could it also be because of naturally low loading capacity of liposome composition chosen in your studies?  Please provide some insights.  Thank you.

7.  In line 300, therefore PEG does not provide advantage to loading capacity of liposomes.  I would recommend authors to highlight this, as many researchers in drug delivery community over-praise and over-state the advantage of using PEG-liposome as a drug carrier.  Thank you.

8.  In line 442 to 443, does this mean that structure of albumin is not affected by the presence of or upon interaction with liposomes?  This is a very interesting observation, as in the past 2 decades, most researchers report findings that protein or peptide undergo structural changes upon interacting with liposomes or lipidic environment.  Does this mean that no structural change of albumin may possibly provide or lead to signals to target liposomes for degradation?  Please provide some insights on this aspect.  Thank you.

9.  It is very good to see that authors put major focuses on detailed micromolecular level changes (e.g., H2O or hydration shell displacement upon interaction with albumin) of phospholipids, i.e., PO2- vs. C=O vs. methylene group on acyl chain.  This provides many important insights from lipid membrane perspective on this biological event.  Please emphasize the importance of your work.  Thank you.

10.  Albumin seems not to insert into lipid membranes until 10% CMG (or 10% PI) is present.  Since CMG (or PI) is not a major component of cell membranes, could 10% CMG (or 10% PI) be a leading/deciding factor player to activate an unknown or known biological event?  For example, could this happen:  "CMGs localize to a region of cell membrane to increase local concentration of CMG to ~ 10% --> trigger bound albumin to insert partially into cell membranes --> lead to formation of small transient pores, although leaky, that allows rapid movement of small molecules or ions across cell membranes"?  Please provide some insights.  Thank you.

11.  In line 467, albumin seems to interact better with negatively charged liposomes.  However, most proteins are negatively charged and thus should interact with positively charged liposomes better.  This albumin interaction seems to give very different insights, even if authors confirmed better albumin interaction with positively charged liposomes in their previous studies, so I encourage authors to shed more lights on this aspect.  Thank you.

12.  In line 601 ~ 604, albumin may denature to reform bonds with bilayer surface.  Does this interaction also apply to other proteins?  Can this be generalized?  If possible, this can be a huge impact on understanding of lipid membrane interaction.  Please emphasize on this important hypothesis, even if subject to future investigation to confirm the validity.  Thank you.

I recommend to publish this manuscript as it is after address my questions.

Author Response

We thank you for your evaluation of our work!

  1. Two different ratios of ganglioside GM1 and CMG-conjugate were studied based on following suppositions: in the works of Yokoyama and colleagues [Yokoyama S. et. al Col.Surf.B, 2003, 27, 141-146; Ohtsuka I. and Yokoyama S. Chem. Pharm. Bull., 2005, 53, 42-47] the optimum concentration of ganglioside in phosphatidylglycerol (PG) liposomes was reported to be 10% and they also reported that at 15 mol % of ganglioside whole liposome surface is covered with sugar moieties. And as PG lipid used by Yokoyama and colleagues has smaller polar head than PC lipid [Lewis R.N.A.H. et. al. BBA, 2005, 1668, 203-214], we decided to make a sample with reduced quantity of the ganglioside with a bulky polar head in the bilayer to minimize steric hindrance if any occurs at 10 mol %. The decrease in GM quantity may also be needed when we prepare liposomes with more than 2 components. For example, when we prepared liposome with the lipophilic prodrug of melphalane the most stable composition in plasma was indeed with 2% of GM1 [Tretiakova D. et.al, Col.Surf.B, 2018, 166, 45-53]. The same logic was used for CMG-conjugate as well.
  2. We have not observed any precipitation of our liposome suspensions in PBS or other buffers used. Although we always work with freshly made samples, we observed that liposomes made with pure PC are stable after extrusion for a couple of weeks. In addition most of our samples are negatively charged and aggregation is not a favorable process for them.
  3. OD540 was used as it is the closest wavelength value on our equipment to the wavelength recommended by the manufacturer. In principle this assay is based on oxidative coupling reaction with 4-aminoantipyrine to yield a quinone-imine dye with absorbance maximum in green to yellow part of spectrum.
  4. Experiments conducted at 25°C were as follows: size, zeta-potential, ANS fluorescence and anisotropy measurements. All these experiments yield biophysical information about our liposomes. As for TMB-PC anisotropy measurements, considering that main lipid in egg PC is POPC with Tm below zero (-2°C, according to Avanti Polar Lipids Inc.), which is 90% of lipids in our samples, we believe that temperature change from 37°C to 25°C and vice versa will not affect TMB-PC signal. Moreover, phosphatidylinositol addition to ePC decreased Tm, the value obtained for 10PI sample in our previous studies was roughly -9°C.
  5. Experiments conducted at 37°C were as follows: determination of protein binding (PB and BSA per liposome values, section 3.2), assessment of the liposome stability during protein adsorption (section 3.5) and ATR-FTIR spectroscopy (section 3.6). These experiments describe different sides of protein-liposome interaction at physiological conditions.
  6. In lines from 295 to 300 (new numbers 308-314) we stated that protein quantities are scarce in comparison with our theoretically calculated range, but earlier in the work of Ohtsuka and colleagues (new lines 291-296) it was suggested that oligosaccharide moieties could shield liposomal surface from protein. But if we take into consideration the results of Kristensen and Yokouchi (references 22 and 27) that albumin tend to bind hydrophobic spots, then the capacity of liposome surface to bind albumin is restricted by the amount of said spots. We believe that during interaction both these factors are important.
  7. We thank you for your advice.
  8. Lines 442 to 443 (new lines 463-464) stated: “Also, there were no significant differences in albumin structure when incubated without liposomes”. Thus, we observe no significant changes in albumin structure when it was incubated alone in buffer for 20 min. This was the control sample as we wanted to be sure that nothing happens in the spectrometer cell without liposomes. In samples containing liposomes we have observed changes in protein structures for two samples 10PI and 10CMG.
  9. We are very pleased to get this kind comment. Thank you very much.
  10. As CMG-conjugate is a part of synthetic neoglycolipid created by our colleagues it is not presented in cell membranes. In liposomes it is difficult to imagine CMG-lipid migration in bilayer to form some kind of separate phase from PC as this molecule has a very distinct negative charge which makes unfavorable CMG clustering. Phosphatidylinositol is a minor membrane component and its derivative PI(4,5)P does migrate to rafts in cell for signal transduction, yet it is a different molecule and right there will be ganglioside in the same raft that does not affect albumin structure. Thus, it is difficult to imagine how albumin will interact with a raft.
  11. This line and Table 2 show albumin needs to displace less water molecules to reach phosphate group on PC surface than on other liposomes. But the amount of tightly bound albumin molecules was the same in our experiment for practically neutral PC liposomes (-12 mV) and for negatively charged 10CMG (-62 mV). Current view on protein adsorption onto liposomes is that charge impacts the quantity of bound protein. And this amount increases in line neutral-negatively charged-positively charged liposomes. Thus, positively charged liposomes bind more protein in relation to protein per liposome ratio and our findings do not contradict with published literature.
  12. From our point of view the ability to unfold in order to interact with bilayer could not be a unique feature just for albumin. It was observed that positively charged cytochrome c unfolds on negatively charged cardiolipin surface and this results in enhanced enzymatic activity of the protein (Muenzner J. et al. J. Phys. Chem. B, 2013, 117,12878-12886.). Although albumin is a very flexible protein, which needs to bind vast majority of ligands, whereas some proteins are very stable and probably would not be affected by bilayer.

Reviewer 2 Report

This manuscript presents a detailed analysis of the interaction of albumin with PC liposomes to which negatively charged lipids have been added. Although it is a small change, it has been detected seriously and accurately. This paper is, therefore, deemed appropriate for publication in this journal.

 -If possible, additional measurements of environmentally responsive fluorescent reagents, such as prodan and laurdan, for membrane hydration, and CD for albumin conformational changes would further confirm the observations. At least, I would like to have comments on the advantages of the present measurement techniques over these methods (in sensitivity or sample preparation).

 -In Figure 8c, the results of deconvolution of each sample should be clearly indicated.

 - p9 “an indication that protein saturation level was already reached somewhere between 10 and 420 molecules per liposome and further increase up to 4550 molecules per liposome does not enhance protein binding“. I did not understand the basis for the numbers in this sentence. I hope it should be stated more clearly.

Author Response

We thank you for your comments on our work!
1. To the best of our knowledge, laurdan probe is mostly used for membrane fluidity measurements as it embeds in the bilayer and its’ emission maximum exhibits redshift when membrane changes its packing from gel to fluid phase. Meanwhile, ANS probe tracks the availability of hydrophobic spots on the outer leaflet of the lipid membrane. In our study the information that could have been obtained from laurdan would be probably similar to BODIPY-lipid anisotropy. In such a this case, BODIPY probe is more favorable to us than laurdan because the former one does not differ from phosphatidylcholine in the polar region. Thus, when we add albumin, we are sure that its binding to our liposomes is not driven by any other functional groups on the surface of the membrane other than choline groups and PI, GM1 or CMG used in the study.

2. For everyones` convinience  we have created publicly available file with C=O peak deconvolutions for each sample at different time points at Zenodo repository adding new reference number [42]. 

3. We have changed the sentence in question and hope that it is easier to understand now.

Reviewer 3 Report

1. Add the paragraph "Statistical analysis of experimental data" to the section of the manuscript "Materials and Methods". What program was used to analyze the statistics?

2. More clearly formulate conclusions.

Author Response

We thank you for your review of our work!

  1. The paragraph "Statistical analysis of experimental data" has been added as subsection 2.8.  All our data presented in tables and graphs were analysed in QtiPlot program.
  2. We made some changes to "Conlusions" and hope that they improved this section.